# Social and Behavioral Pathways between Adverse Childhood Experiences and Poor Adult Physical Health: Mediation by Early Adulthood Experiences in a Low-Income Population

**DOI:** 10.3390/ijerph191710578

**Published:** 2022-08-25

**Authors:** Claire Devine, Hannah Cohen-Cline

**Affiliations:** Providence Center for Outcomes Research and Education, Portland, OR 97213, USA

**Keywords:** adverse childhood experiences, intimate partner violence, substance use, social isolation, employment, mediation

## Abstract

Adverse childhood experiences (ACEs) increase the risk of poor health and wellbeing in adulthood. In this study, we tested whether experiences in early adulthood—intimate partner violence (IPV), substance use, social isolation, and work instability—mediate the relationship between ACEs and poor physical health in later adulthood. Using data from a large-scale survey of Medicaid enrollees in the Portland metropolitan area, four separate mediation models were constructed to assess the indirect effects of each early adulthood experience and the proportion of the total effect on physical health accounted for by the pathway. Experiencing four or more ACEs increased the risk of poor adult physical health by 50% (RR 1.50). Considered in separate models, mediation by IPV accounted for 14.4% of the total effect; substance use mediated a similar proportion (14.0%). Social isolation was a less substantial mediator (7.6%). Work instability did not mediate the relationship between ACEs and adult physical health in our population. These findings provide evidence that IPV, substance use, and social isolation in early adulthood are part of the pathway between high ACEs and poor adult physical health. Intervening to prevent negative early adult experiences may mitigate some of the long-term effects of childhood trauma on health.

## 1. Introduction

Adverse childhood experiences (ACEs), defined here as experiencing abuse, neglect, or household dysfunction during childhood, have substantial effects on adult well-being. The original ACEs study, conducted by Felitti et al. in 1997 in adult members of a large health management organization, found a graded relationship between number of ACEs and the leading causes of adult death, including heart disease, cancer, chronic lung disease, and liver disease [1]. Since then, ACEs have been linked to increased likelihood of negative health outcomes in adulthood, including self-reported health, functional limitations, metabolic syndrome, and ischemic heart disease [2,3,4,5,6,7].

Several studies have examined the mechanisms by which abuse and trauma during childhood affect adult health, with attention given to biological causes such as altered or disrupted neurodevelopment and brain chemistry, stress physiology, and epigenetic effects [8,9,10,11]. Allostatic load and cumulative life stress may also contribute to the relationship between maltreatment and poor health [12,13,14,15].

However, research has also focused on the mediation of the effect on physical health by increased risk behaviors and, slightly less frequently, life course experiences. Unlike altered neurodevelopment, life course experiences are of particular interest because they may be more easily modifiable.

The life-course view of childhood adversity suggests that early traumatic experiences are part of “life course cascades” [16] that have the potential to lead to additional stressful experiences that accumulate over an individual’s life and negatively affect their health. In the life-course view, any experience or event in that cascade is a potential place for intervention.

While there is limited previous research on pathways leading from ACEs to poor physical health, studies have established a link between ACEs and many social and behavioral experiences that may impact health in adulthood. Childhood adversity has been consistently and significantly associated with an increase in intimate partner violence [17,18], social isolation [19], substance use [20,21,22], and unemployment [23,24]. Existing studies that test these social and behavioral experiences as mediators often focus on mental health or social functioning-related outcomes. In particular, intimate partner violence and social isolation have been shown to increase the risk for poor mental health, including depression and suicidality [18,25,26,27,28,29,30]. Substance use has mediating effects on suicidality [31] as well as legal-related outcomes [32,33].

A few studies have examined the mediating relationship of social and behavioral pathways between ACEs and physical health. Low-income status and socioeconomic factors have been found to be mediators between ACEs and physical health [34,35]. Community and social support may also be part of the pathway by serving as protective factors against lower physical health quality [36,37]. A study by Seon et al. on intimate partner violence also showed partial mediation with physical health in college students [18]. However, there remain gaps in the literature on mediation by life-course experiences with a focus on adult physical health. There is a particular need for this analysis on physical health in low-income populations, which studies have shown to have a high burden of ACEs [38,39].

### Current Study

Despite studies showing that adults with a lower socioeconomic status have a disproportionate burden of ACEs, few studies of the social and behavioral mechanisms by which ACEs lead to poor adult physical health have been conducted specifically in low-income populations. For this reason, our study focused on adults enrolled in Medicaid, a government-funded healthcare insurance program for low-income individuals in the United States. The purpose of this study was to quantify the mediating effects of four social and behavioral pathways in early adulthood: intimate partner violence, substance use, social isolation, and trouble keeping steady work. We hypothesized that these four experiences would partially mediate the association between ACEs and self-reported poor physical health in middle to late adulthood.

## 2. Materials and Methods

### 2.1. Survey Design

The study leveraged a large-scale survey fielded in 2015–2016 that gathered information on patterns of life-course adversity and support, as well as health and healthcare outcomes in adulthood. Claims data from Medicaid-enrolled adults in the Portland, OR Metropolitan Area were used to identify a representative sample for the survey, with oversampling of key populations—including members with high health care utilization, members with high medical complexity, and Black/African American members—to ensure adequate sample sizes for subgroup analyses. High health care utilization was defined based on multiple emergency department visits and/or inpatient stays per member per year. High complexity members had projected health care costs in the top 10% of the sampling universe, which we determined using the Chronic Illness and Disability Payment System (CDPS Plus) risk score algorithm, which is a validated risk scoring algorithm that uses diagnoses and prescription data from claims data to estimate projected costs [40]. The survey was fielded through the mail, with intensive outreach via follow-up phone calls and in-person visits for a random sample of the non-respondents. Survey questions related to a participant’s experiences during childhood and teenage years (0–18 years), young adulthood (19–30 years), and later adulthood (31 years or older).

### 2.2. Participants

Survey responses were collected for 2385 individuals (response rate 26%) enrolled in Medicaid, a government healthcare assistance program for low-income individuals in the United States. In mediation analysis, all mediators must follow the exposure and precede the outcome in time. Because the mediators in the current analysis were derived from questions about experiences occurring between 19 and 30 years of age, and the outcome was current physical health, the eligible sample was limited to participants who were at least 31 years of age. The study was further restricted to participants who had complete information on: ACEs score, self-reported physical health, age, gender, and the four potential mediators. Although each mediator was assessed separately, we used the same eligible population in each of the four models to give a general sense about the relative strength of the pathways across the independent models. A total of 1893 participants were included in the analysis.

### 2.3. Measures

#### 2.3.1. Adverse Childhood Experiences (ACEs)

Respondents to the survey were asked whether they experienced verbal, physical, or sexual abuse; physical or emotional neglect; or any of the following measures of household dysfunction: living with a household member who had a mental illness, living with a household member who had a substance use disorder, living with a household member who had been to jail or prison, violence between caregivers, or parental divorce/separation.

Consistent with previous literature, we calculated the ACE score as the count of ACEs a participant experienced at any time between 0 and 18 years of age [6,41]; scores ranged from 0 (none of the events occurred) to 10 (all the events occurred). The score was further collapsed into a binary variable for the purposes of the analysis, based on whether a participant had four or more ACEs (“high ACEs score”) or fewer than four ACEs (“low ACEs score”), which is commonly used as a cut-off for binary ACEs measures [1,6,23,42]. The Cronbach’s alpha for the ACEs variable was 0.83, which shows good reliability for the scale.

#### 2.3.2. Self-Reported Physical Health

A 5-point Likert scale was used for participants to rate their current physical health. Responses were recategorized as a binary variable, with participants who indicated that they had “poor” or “fair” physical health included in one category, and participants who felt they had “good,” “very good,” or “excellent” physical health in the other. Prior research has shown that the reliability of single-item physical health questions such as this one are valid predictors of illness and mortality and perform as well as multi-item scales [43].

#### 2.3.3. Potential Mediators

Respondents indicated whether or not they experienced certain life course events from 19 to 30 years old (”early adulthood”). Four types of experiences during this time period were considered as potential pathways between ACEs and health in later adulthood. All mediators were treated as binary variables.

Intimate partner violence (IPV): reporting experiencing physical abuse from a partner or loved one, emotional abuse from a partner or loved one, or sexual abuse.Substance use: reporting being a problem drinker, alcoholic, or user of street drugs.Social isolation: reporting no close relationships to people they could count on.Work instability: reporting not being able to find and keep steady work.

These mediators were proposed based on a literature search of life course events that may be linked to childhood adversity [20,44,45,46,47,48,49] and that have the potential to affect physical health in adulthood. These experiences not only made sense in our conceptual models but also fit our focus on identifying potentially modifiable mediators. IPV was the only mediator constructed from more than one item; the three items in the scale had a Cronbach’s alpha of 0.68; while this suggests slightly lower reliability than the more accepted cut-off of 0.7, it is likely due in part to the low number of items in the scale.

#### 2.3.4. Covariates

We decided a priori to include age and gender in the four mediation models because of possible confounding effects on the exposure-outcome, exposure-mediator, and mediator-outcome relationships [50,51,52]. Age was derived from an individual’s Medicaid claims data. Information on gender was self-reported as male or female on the survey. Additional covariates were not included due to model convergence issues.

### 2.4. Statistical Analysis

Mediation analyses allow us to examine the potential mechanisms by which an exposure (four or more ACEs) affects an outcome (poor adult physical health). Where mediation is present, the exposure causes the mediating experience, which causes the outcome—in our case, we hypothesized that each proposed mediator would only partially mediate the association. We built four separate mediation models to estimate the effects of each mediator and the proportion of the total ACEs—physical health association that operated through the hypothesized pathway (Figure 1).

We used pairwise correlations and regression models to determine whether each pathway met the underlying requirements for mediation: (1) the exposure is significantly related to the mediator, and (2) the mediator is significantly related to the outcome measure. Given the continuing and complicated nature of trauma, we also accounted for potential interactions between each mediator and the exposure. The interaction between ACEs and the mediator was tested for all models; interaction terms significant at *p* < 0.10 were retained in the model when assessing the indirect and direct effects.

Analyses followed the Valeri and VanderWeele counterfactual approach for assessing controlled and natural direct effects, natural indirect effects, and total effects of the mediators, based on log-linear models [53].

Our focus was on the natural indirect effect of each mediator, which assesses the amount the outcome would be affected, with the exposure controlled as present (four or more ACEs), when the mediator is changed from the level it would take in the absence of the exposure to its level in the presence of the exposure [53].

We also calculated the controlled direct effect, which controls the mediator at a uniform level in the population and looks at the change in the outcome in the presence versus the absence of the exposure, and the natural direct effect—which controls the mediator at the level it would take for each individual in the absence of the exposure and looks the change in the outcome when the exposure is present compared to when it was absent [53]. When there is no interaction between the exposure and mediator, the controlled and natural direct effects are the same.

The total effect calculates how much the outcome changes based on the presence or absence of the exposure and decomposes into the direct and indirect effects [53]. The proportion mediated for a binary outcome transforms the odds ratios to assess the ratio of the natural indirect effect to the total effect [53].

Given the high prevalence of the health outcome and of each of the four mediators in the population, we followed the recommendations of Valeri and VanderWeele (2013) and built our analyses on a risk ratio rather than odds ratio scale [53]. Regression models incorporated survey weighting based on the sampling scheme; bootstrapping was used to calculate standard errors. Confidence intervals are not provided for the proportion mediated as this estimate tends to be unstable [53]. All log-linear models were built with R.

## 3. Results

Slightly less than half of all adults in the eligible study population had experienced four or more ACEs (Table 1). The two groups were similar in terms of age; however, those with high ACE scores were slightly more likely to be female, Hispanic, and Black/African American. They were also more likely to report poor or fair physical health, compared to individuals with fewer ACEs.

The most prevalent mediator was IPV; we estimated that 61.5% of the total population had experienced IPV. The prevalence of IPV in adults with high ACEs was nearly twice that of adults with low ACEs (78.8% versus 40.0%). Social isolation was the least prevalent mediator in the population and had the smallest difference in prevalence between the high ACEs and low ACEs group (25.7% versus 12.4%). Experiencing four or more ACEs increased the risk of poor physical health in late adulthood by 50% (RR 1.50, 95% CI: 1.24–1.82).

Table 2 shows the result of the mediation analyses. Of the four models, only the interaction term between ACEs and work instability was significant (*p* = 0.03) and was included in the model.

Because our goal was to assess the strength and significance of each proposed mediator, we focused on indirect effects and the proportion mediated. We found that IPV, substance use, and social isolation each significantly mediated a portion of the relationship between high ACEs and poor health (*p* < 0.05 for each model). The strongest mediators were IPV and substance use. The risk ratios for both the IPV and substance use indirect effects were 1.05 (95% CI: 1.01–1.10 for IPV; 95% CI: 1.02–1.09 for substance use). We estimated that each of these early adulthood experiences, assessed in separate models, mediated approximately 14% of the high ACEs—poor physical health relationship.

Social isolation was also a significant, although comparatively weaker, mediator. The risk ratio for the indirect effect of this pathway was 1.03 (95% CI: 1.01–1.06), and the proportion mediated was estimated as 7.6%.

We did not find evidence of mediation between high ACEs and poor health by work instability.

## 4. Discussion

The goal of this study was to explore mechanisms by which ACEs lead to poor physical health in later adulthood. Consistent with our expectations, we found that participants in our study population with four or more ACEs were more likely to have fair or poor physical health in adulthood than those with fewer than four ACEs. Of the four life-course events that we predicted would be pathways between ACEs and poor health, three were significant mediators: intimate partner violence, substance use, and social isolation. Of these, IPV and substance use had the most substantial indirect effects. We did not find evidence that the last hypothesized mediator—work instability—mediated the ACEs-adult health relationship.

One of the strengths of our study is that it was conducted in a Medicaid-enrolled population, which is largely low-income. Although studies have shown that low-income populations have a higher burden of ACEs [38,39], there has been less research to understand the impacts of ACEs on physical health in this population. Some studies on ACEs and physical health outcomes have considered socioeconomic status as a potential mediator [34,35], as discussed below. While these studies assessed measures of socioeconomic status as mediating pathways, the advantage of our work is that it both tests the mediating effect of a specific experience tied to socioeconomic status (work instability) and gives insight into how social and behavioral mechanisms function within the context of a low-income population.

Efforts to intervene will have the greatest impact in populations with a high burden of ACEs, and it is important to study the mechanisms by which ACEs affect adult health in those populations, specifically. The Medicaid population is of particular interest in studying social and behavioral pathways between ACEs and physical health because researchers have suggested using Medicaid initiatives to focus on life-course health development [54] and utilizing accountable care organizations to promote family-centered care that may prevent or mitigate adverse childhood experiences [55]. This analysis helps define life course events that could be the focus of such interventions.

An important contribution of this study to previous research is the finding around the indirect effects of intimate partner violence. Previous studies have shown that, in addition to ACEs, IPV may be more prevalent in low socio-economic populations [56], and have established the association between ACEs, IPV, and multiple physical health consequences such as chronic pain, gastrointestinal disorders, and chronic disease [44,57]. However, a limited number of studies have looked in detail at potential pathways between ACEs and poor physical health through early adulthood IPV. A study by Seon et al. found partial mediation of physical and mental health by IPV in college students [18]. Another study showed that relationship distress significantly mediated the relationship between ACEs and adult physical and mental health, but defined distress in terms of relationship satisfaction and difficulty and did not explicitly consider IPV [58]. A study by Willie et al. found that, for adult women currently experiencing IPV, those with a high ACEs profile were significantly more likely to report depressive and PTSD symptoms, but did not assess the effect on physical health nor consider IPV as a mediator [59].

Given the number of potential pathways between childhood maltreatment and poor health, identifying an early adulthood experience that accounts for 14% of the association is an important step in developing targeted interventions that mitigate the effects of ACEs on health. The National Intimate Partner and Sexual Violence Survey (2015 update) estimated that around 43.6 million women (36.4%) and 37.3 million men (33.6%) in the United States experienced IPV during their lifetime [60]. Despite this high prevalence, IPV interventions are typically skewed toward response, rather than prevention [61]. Most prevention programs in high-income countries are school-based group trainings, which have varying levels of effectiveness but may have cumulative effects [61]. Policy changes such as the Violence against Women Act have been credited with past reductions in intimate partner violence rates within the United States [61]. This research provides evidence of the importance of strengthening prevention programs and ensuring the continuation of policies that protect and support victims of IPV.

Substance use accounted for an equally large proportion of the pathway (14%) between ACEs and poor physical health. The increase in the prevalence of substance use in adulthood for those who experienced childhood trauma has been well-established [1,20,46,50]. Studies on the role of substance use as a mediator usually focus on the relationship between ACEs and mental health outcomes such as suicide attempts [62], or legal-related experiences such as gang involvement [33] and recidivism [32]. This study builds on previous literature by focusing more specifically on mediation of physical health outcomes. However, substance use is often a coping mechanism for trauma [63] and frequently begins in childhood or adolescence [20]; this may complicate the timing of trauma-informed interventions to prevent this pathway, which are most successful before substance use becomes chronic [63].

It is important to understand the nuances around substance use when designing interventions. Research on substance use programs has demonstrated substantial differences between adolescents or young adults and older adults that may affect the efficacy of interventions, in part due to physiological changes that affect emotion regulation and risk taking, and life-course milestones such as ending education or career transitions [64]. Programs and policies therefore need to be designed explicitly for adolescents and young adults. Likewise, policies around taxation, minimum legal ages, and availability have been shown to be effective in preventing alcohol abuse, but cannot be applied to illicit drugs, which still need innovative approaches to prevention [64].

The findings around social isolation are supported in the literature as well. Herrenkohl et al. used structural equation modeling to show that the presence of safe, stable, and nurturing relationships in childhood mediated the relationship between childhood abuse and both general self-reported health and somatic complaints in adulthood [37]. Our analysis builds on previous literature by specifically looking at social isolation in early adulthood, rather than during childhood. In our study, social isolation was the least prevalent mediator in the population. However, the overall increase in social isolation and loneliness during the COVID-19 pandemic [65] has likely increased the number of individuals that may be affected by this pathway. This mechanism may also warrant reexamining in the future, to determine if the mediating effects have compounded or otherwise changed as a result of the pandemic.

Our results for IPV, substance use, and social isolation are largely consistent with prior research; however, there are some differences in our lack of findings regarding work instability and previous research into socio-economic status. Socioeconomic status as a construct is often represented in research through multiple proxy variables, including income, education, and employment. Other studies have found there is a significant, if modest, indirect pathway between ACEs, low income, and poor physical health [35]. Similarly, Font et al. (2016) found that various socioeconomic conditions such as income and education accounted for a significant proportion of the association between ACEs and physical health [34]. Our current study differed from these in two key ways that might account for the contrasting results. First, we used a different measure of socioeconomic status (income and education vs. work instability), which may have different impacts on physical health. Second, we focused on a generally low-income Medicaid population, and it may be that for this population work instability in young adulthood is a less important pathway.

Our findings suggest the “second hit” of negative experiences during early adulthood in individuals with multiple ACEs are important pathways by which ACEs lead to adverse health consequences. The evidence supports the idea that there may be effective interventions in early adulthood that can ameliorate the long-terms effects of childhood trauma on adult physical health. These results further emphasize the importance of health care providers addressing adverse experiences in their patient panel through a variety of methods. Trauma-informed care practices can prevent re-traumatization, validate and recognize traumatic events, and provide coping strategies [66,67,68]. Cognitive behavioral therapy interventions have been shown to improve social, cognitive, and emotional functioning and reduce health-risk behaviors in adults with ACEs [69], which may help prevent some of the life-course experiences that mediate the relationship with poor adult physical health. The health care setting is one logical place to identify young adults with a high number of ACEs; knowing a young adult’s history of trauma, in combination with their current life experiences, will give providers additional information about their patient’s risk of poor physical health outcomes, and create opportunities to mitigate this harm.

### Limitations

This study has some limitations. First, the extent to which our findings can be generalized is primarily limited to a white, urban population. While Black/African American individuals were oversampled, the study population was still predominantly white, and only English and Spanish versions of the survey were distributed. This is generally reflective of the heavily white population in the Portland Metropolitan Area but does limit the generalizability of the results.

The use of surveys also introduces the possibility of response or recall bias; difficulties with recall may have been particularly relevant because the study participants, all of whom were 31 years or older, were asked to remember experiences that occurred during childhood. However, previous studies have found that the individuals can reliably recall childhood trauma, regardless of age or current mental state [70,71].

Additionally, the ten-item ACEs scale has be critiqued for not including other types of trauma, such as the experience of discrimination or bullying [42]. In addition to limiting the definition of adversity in childhood, the exclusion of experiences of discrimination is a particular limitation as these experiences may be much more relevant in populations already facing health and economic disparities. Future studies should expand beyond the ACEs scale to assess the impacts of racism and discrimination on participants and include subgroup analyses to consider how the effects may be moderated by race, ethnicity, or other population.

There were some limitations associated with our statistical approach. Our analysis assumes no uncontrolled confounding in the underlying regression models. While we controlled for age and gender, additional covariates caused convergence issues for the models, and excluding them may have affected our results. The definition of gender in our study is further limited because the survey question responses were binary, only allowing for “male” or “female.” Non-binary or transgender individuals may therefore have been excluded from the study because the gender question did not have an applicable response option for them. In addition, some of the mediating experiences tested in this study are more common in one gender versus another, such as IPV [60]. Future research should explore mediation effects separately, by gender, to understand for whom these mediators have the greatest impact.

There may also be temporality issues in the model; proposed mediators are assumed to occur after the ACEs exposure, but some—particularly substance use [20]—may have begun in adolescence or earlier and could potentially predate the childhood experiences. In our analysis, approximately 70% of participants with substance use in early adulthood also reported substance use in adolescence. In addition, the proposed mediator variables are highly correlated with each other; considering all four potential mediators in the same model would parse out which pathways are the strongest in the presence of the others. Future studies should expand this approach to structural equation modeling, to allow for the inclusion of all four mediators.

## 5. Conclusions

While it is well-established that experiencing trauma, abuse, and neglect in childhood can have negative effects on health later life, this study adds to the growing body of work exploring the mechanism by which that harm occurs. Our results provide evidence that relationship between ACEs and poor adult health is partially mediated by subsequent negative experiences in early adulthood in a low-income population. Identifying modifiable pathways through which ACEs impact adult health will inform what interventions may mitigate the consequences of ACEs.

## Figures and Tables

**Figure 1 ijerph-19-10578-f001:**
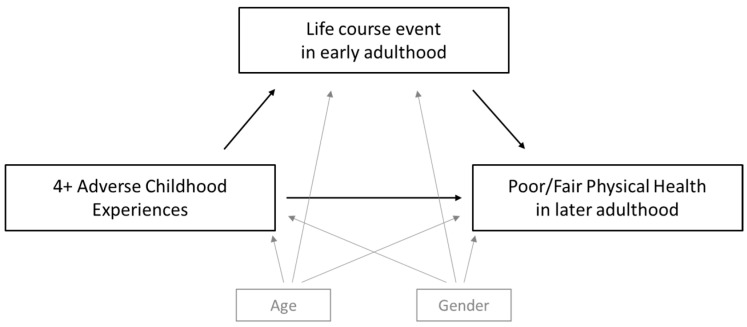
Directed acyclic graph of mediation model.

**Table 1 ijerph-19-10578-t001:** Characteristics of study population.

	<4 ACEs 58.1% (*n* = 928)	4 + ACEs 41.9% (*n* = 935)
Poor or Fair Physical Health	32.2% (452)	48.4% (606)
Age (mean (SE))	48.9 (0.56)	47.4 (0.62)
Female	58.7% (523)	64.7% (622)
Hispanic	6.2% (54)	8.7% (71)
Black/African American	5.2% (133)	7.3% (116)
**Early-Adulthood Mediators**		
IPV	40.0% (417)	78.8% (777)
Substance use	21.5% (262)	51.0% (520)
Social isolation	12.4% (104)	25.7% (296)
Work instability	20.3% (206)	36.6% (350)

Percentages are survey weighted to estimate population values. Number of observations apply to the eligible sample.

**Table 2 ijerph-19-10578-t002:** Adjusted risk ratios ^1^ for direct and indirect effects and proportion mediated.

	IPV ^2^	Substance Use ^2^	Social Isolation ^2^	Work Instability
Controlled Direct Effect	1.34 (1.10–1.66)	1.34 (1.10–1.67)	1.42 (1.16–1.75)	1.08 (0.80–1.49)
Natural Direct Effect ^3^	1.34 (1.10–1.66)	1.34 (1.10–1.67)	1.42 (1.16–1.75)	1.51 (1.24–1.86)
Natural Indirect Effect	1.05 (1.01–1.10)	1.05 (1.02–1.09)	1.03 (1.01–1.06)	1.01 (0.99–1.03)
Total Effect	1.41 (1.17–1.72)	1.41 (1.17–1.73)	1.46 (1.21–1.79)	1.52 (1.26–1.87)
Proportion Mediated	14.4%	14.0%	7.3%	1.6%

^1^ Models adjusted by age and gender; effects calculated for females, 50 years old. ^2^ Indirect effect significant at *p* < 0.05. ^3^ For models without an exposure-mediator interaction term, CDE and NDE are the same.

## Data Availability

The survey data presented in this study are available on request from the corresponding author, with appropriate Ethics Committee or Institutional Review Board review and approval. Restrictions apply to the availability of the claims data. Data was obtained from Health Share of Oregon and are available with permission of Health Share of Oregon (https://www.healthshareoregon.org/, accessed on 1 April 2018).

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
