# Peer review of "Social and Behavioral Pathways between Adverse Childhood Experiences and Poor Adult Physical Health: Mediation by Early Adulthood Experiences in a Low-Income Population"

_ijerph, 2022, doi:10.3390/ijerph191710578_

Round 1

Reviewer 1 Report

The current study used a regional sample of adults to determine what mediated the association between adverse childhood experiences and poor adult physical health. The authors report such an association can be mediated by IPV, substance use, and social isolation. While this is an interesting study, there is room for improvement before publication. I outline my comments and questions below in the hope that the authors can clarify or explain them to help improve this study.

I would suggest the title be revised as the article analyzed four early adulthood experiences: intimate partner violence (IPV), substance use, social isolation and job instability. In my opinion, it is not accurate enough to use 'social pathways' to summarize these factors (such as substance use). It would be better if they could be summed up in another term.

Insufficient review of the 4 mediators involved in the study in the 'Introduction' section, e.g., any research on intimate partner violence related to this topic? An exhaustive review of previous studies is recommended as the basis for this paper and to show how this paper makes modifications to the research gaps.

The first paragraph in the 'Materials and Methods' section comes from the original template that should be removed.

What is the response rate to the survey? It should be reported as a part of the survey methodology.

Is there a limited period for the measurement of each potential mediator? For example, in the last 12 months or lifetime? Clarification is recommended.

One issue is not very clear throughout the text: was this study conducted among a low-income population? (It was mentioned in some places.) Was the sample of this survey from low-income people? Or the entire population? Clarification is suggested to be made in the full text. 

If the study was aimed at the low-income population, the 'Discussion' section should be more targeted (written more specifically) according to the situation of low-income people. Similarly, it would be better if this population had been identified in the title.

When was this data collected? Looks like a cross-sectional survey?

The data for childhood experiences are retrospective, so why not use OR? Currently RR is used, but it does not appear appropriate for this data.

I would like to see some policy recommendations and appeals for IPV and substance use, which would be valuable for promoting practice.

Overall, this study poses interesting questions with some potential mediating factors examined. If these issues can be improved, I recommend publishing this article.

Author Response

We would like to thank the reviewers for their insightful comments on our manuscript. We have updated the manuscript to incorporate many of their recommendations. We have provided detailed responses to each of their concerns below.

Reviewer 1

The current study used a regional sample of adults to determine what mediated the association between adverse childhood experiences and poor adult physical health. The authors report such an association can be mediated by IPV, substance use, and social isolation. While this is an interesting study, there is room for improvement before publication. I outline my comments and questions below in the hope that the authors can clarify or explain them to help improve this study.

  1. I would suggest the title be revised as the article analyzed four early adulthood experiences: intimate partner violence (IPV), substance use, social isolation and job instability. In my opinion, it is not accurate enough to use 'social pathways' to summarize these factors (such as substance use). It would be better if they could be summed up in another term.

We have updated the manuscript title to account for both social and behavioral pathways. We have also specified the population for the study, as recommended by the reviewer in Comment #7.

  1. Insufficient review of the 4 mediators involved in the study in the 'Introduction' section, e.g., any research on intimate partner violence related to this topic? An exhaustive review of previous studies is recommended as the basis for this paper and to show how this paper makes modifications to the research gaps.

We would like to thank the reviewer for their comment. We have expanded the Introduction section to include additional information on the link between ACEs and social and behavioral life experiences, as well as their role as mediators of health and other outcomes.

  1. The first paragraph in the 'Materials and Methods' section comes from the original template that should be removed.

We would like to thank the reviewer for their comment and the opportunity to correct this issue.

  1. What is the response rate to the survey? It should be reported as a part of the survey methodology.

We have added this information on response rate in the Materials and Methods section, in 2.2 Participants.

  1. Is there a limited period for the measurement of each potential mediator? For example, in the last 12 months or lifetime? Clarification is recommended.

We would like to thank the reviewer for their comment. The information on time period is found in Materials and Methods in Section 2.3.3 Potential Mediators. We have edited the sentence to increase clarity around this point.

  1. One issue is not very clear throughout the text: was this study conducted among a low-income population? (It was mentioned in some places.) Was the sample of this survey from low-income people? Or the entire population? Clarification is suggested to be made in the full text. 

We would like to thank the reviewer for their comment. The sample of the survey was effectively from low-income people. All participants were Medicaid members, which is a government-funded healthcare program for low-income individuals in the United States. We have added this information throughout the manuscript in the Introduction, Materials and Methods, and Discussion sections to provide some context around the Medicaid program and make it clearer that the population is low-income because of this sample’s inclusion criteria.

  1. If the study was aimed at the low-income population, the 'Discussion' section should be more targeted (written more specifically) according to the situation of low-income people. Similarly, it would be better if this population had been identified in the title.

We would like to thank the reviewer for their comment. We have expanded the Discussion section to address the importance of this study for low-income populations and included information into relevant previous literature. We have also clarified the title of the manuscript so that the Medicaid population is referenced.

  1. When was this data collected? Looks like a cross-sectional survey?

Yes, the survey was cross-sectional, although it allowed for a mediation analysis because questions specified the age ranges in which the exposure, mediators, and outcome occurred, as indicated in section 2.2 Participants. We have added information to the same section to indicate the years when the survey was fielded.

  1. The data for childhood experiences are retrospective, so why not use OR? Currently RR is used, but it does not appear appropriate for this data.

We would like to thank the reviewer for their comment, however we respectfully disagree that risk ratio is not appropriate for the data. Because our study populations were defined by exposure, rather than outcome as in a case-control study, using the risk ratio is appropriate. Using RR also avoids the potential for misinterpretation of the magnitude of the effect introduced by using odds ratios in situations when the outcome is not rare. In our case, both the health outcome and each of the four mediators had a high prevalence. We have added in 2.4 Statistical Analysis our source for the recommendation of RRs in order to add clarity.

  1. I would like to see some policy recommendations and appeals for IPV and substance use, which would be valuable for promoting practice.

We would like to thank the reviewer for their comment and agree that this is a good addition to our manuscript. We have added potential policy implications and recommendations to the Discussion section specific to IPV and substance use, in the paragraphs starting on lines 291 and 315, respectively.

  1. Overall, this study poses interesting questions with some potential mediating factors examined. If these issues can be improved, I recommend publishing this article.

We would like to thank the reviewer for the positive comments on our manuscript.

Reviewer 2 Report

The idea expressed in this manuscript is nice and the authors have trialed it with 2385 participants. However, the authors might pay attention to the followings:

1 The Introduction is a little bit short, the authors might expand their hypotheses based putting up more rationales for the social pathway strains.

2 The authors might isolate their participants into groups such as poor physical health (detailed or combined) and the healthy controls, to figure out the early family or other social factors which might contribute to the later years’ healthy outcomes.

3The authors might notice the adverse and positive life events in the development of physical (un)health of the participants, these social factor contributions are crucial.

4 The authors might also calculate the mediation effects of some particular variables, even though this is a cross-sectional study.

5 The measure of physical health should be a structure-validated method, therefore, the authors might supply the related internal alphas of this measure and other measures such as ACE, and IPV.

6Gender (sex) effects might also be questioned further, using the available data.

Author Response

We would like to thank the reviewers for their insightful comments on our manuscript. We have updated the manuscript to incorporate many of their recommendations. We have provided detailed responses to each of their concerns below.

The idea expressed in this manuscript is nice and the authors have trialed it with 2385 participants. However, the authors might pay attention to the followings:

  1. The Introduction is a little bit short, the authors might expand their hypotheses based putting up more rationales for the social pathway strains.

We would like to thank the reviewer for their comment. We have expanded the Introduction section to include additional information on the focus of the manuscript on life course pathways, as well as additional literature on the relationship between ACEs and social and behavioral life experiences.

  1. The authors might isolate their participants into groups such as poor physical health (detailed or combined) and the healthy controls, to figure out the early family or other social factors which might contribute to the later years’ healthy outcomes.’

We thank the reviewer for this comment. However, we chose to approach this study from a confirmatory analysis perspective, with our exposure, outcome, and potential mediators selected a priori rather than relying on an exploratory and more data-driven approach. Further, as our groups are defined by the exposure (ACEs) and not the outcome (physical health), a case-control study would not be appropriate here. We agree this may be an important step to generate new hypotheses for future studies.

  1. The authors might notice the adverse and positive life events in the development of physical (un)health of the participants, these social factor contributions are crucial.

We thank the reviewer for this comment and agree that understanding the contributions of social factors to the creation of physical (un)health is crucial, which is why we chose to explore the pathways that lead from adverse events in childhood to poor health in later adulthood. We have elaborated on our motivations for these analyses in the Introduction section.

  1. The authors might also calculate the mediation effects of some particular variables, even though this is a cross-sectional study.

We thank the reviewer for this comment. We have included the mediation effects (i.e. natural indirect effects) in Table 2, and discussed the magnitude of these texts and the proportion mediated in the Results section.

  1. The measure of physical health should be a structure-validated method, therefore, the authors might supply the related internal alphas of this measure and other measures such as ACE, and IPV.

We agree that this information is important to include. We have included Cronbach’s alpha measures for the ACEs variables and the IPV variables in section 2.3 Measures. Because our physical health measure was a single item, an internal alpha would not be applicable. However, we added support for the validity of a single-item self-reported health measure in the same section.

  1. Gender (sex) effects might also be questioned further, using the available data.

We would like to thank the reviewer for their comment. We agree that exploring the moderating effects of gender would add value to our study and acknowledge that the pathways assessed could be different across genders. However, we were unable to include this in our analyses because splitting our data into two models led us to concerns around convergence in the model. We have added information into section 4.1 Limitations to call attention to this issue and note that it is an important area for future research.

Round 2

Reviewer 1 Report

I personally suggest changing the title from "in a Medicaid population" to "in a low-income population", which would help the article be searched and found by other researchers, thereby increasing exposure and citations.

Overall, my other comments have been satisfactorily incorporated, and the manuscript has been greatly improved. Support acceptance in the present form.

Author Response

Reviewer comment: I personally suggest changing the title from "in a Medicaid population" to "in a low-income population", which would help the article be searched and found by other researchers, thereby increasing exposure and citations.

We thank the reviewer for this comment and have changed the title accordingly.

Overall, my other comments have been satisfactorily incorporated, and the manuscript has been greatly improved. Support acceptance in the present form.

Reviewer 2 Report

No further comments.

Author Response

Reviewer comment: No further comments.

We thank the reviewer for their review of the paper and thoughtful feedback on the previous round.